# SoftAdaClip: A Smooth Clipping Strategy for Fair and Private Model Training

## Abstract

Differential privacy (DP) provides strong protection for sensitive data, but often reduces model performance and fairness, especially for underrepresented groups. One major reason is gradient clipping in DP-SGD, which can disproportionately suppress learning signals for minority subpopulations. Although adaptive clipping can enhance utility, it still relies on uniform hard clipping, which may restrict fairness. To address this, we introduce **SoftAdaClip**, a differentially private training method that replaces hard clipping with a smooth, $\tanh$-based transformation to preserve relative gradient magnitudes while bounding sensitivity. We evaluate SoftAdaClip on various datasets, including MIMIC-III (clinical text), GOSSIS-eICU (structured healthcare), and Adult Income (tabular data). Our results show that SoftAdaClip reduces subgroup disparities by up to **87%** (average reduction **56%**) compared to DP-SGD, and up to **48%** (average reduction **22%**) compared to Adaptive-DPSGD, with these improvements being statistically significant. These findings underscore the importance of integrating smooth transformations with adaptive mechanisms to achieve fair and private model training.

## 1 Introduction

Machine learning (ML) models are increasingly being used in sensitive fields such as healthcare, finance, and social services, where privacy and fairness are critical. In such applications, models are often trained on data that includes sensitive personal information, such as health records, income levels, and demographic characteristics. Although this information improves the accuracy of predictions, it also introduces significant risks related to biased outcomes and privacy attacks. These risks include membership inference and data extraction attacks, especially in large language models that can inadvertently memorize sensitive training data and expose it through their generated outputs (Li et al., 2021; Carlini et al., 2021; Shokri et al., 2017; Hayes et al., 2017).

To mitigate these risks, several privacy-enhancing technologies (PETs) have been developed (Esipova et al., 2022), including differential privacy (DP) defined by Dwork et al. (2006), federated learning (Sadilek et al., 2021; Ali et al., 2022), secure multiparty computation, and homomorphic encryption.

Among these methods, DP offers formal, mathematical privacy guarantees. An algorithm $M$ satisfies $(\epsilon,\delta)$-differential privacy if the likelihood of its output remains nearly the same, regardless of whether a single data point is included or removed. This is defined as:

$$P[M(S) \in E] \leq e^\epsilon P[M(S') \in E] + \delta,$$

where lower values of $\epsilon$ and $\delta$ correspond to greater privacy protection (Dwork et al., 2006). DP reduces the risk of exposing sensitive information by limiting how much an individual data point can influence the output. This ensures that the presence or absence of any individual has a limited effect on the output, thereby protecting personal information.

DP can be implemented through gradient perturbation-based methods (Abadi et al., 2016; Li et al., 2021), such as DP-SGD and its extensions such as as DP-Adam. These techniques involve three primary steps: (1) subsampling data, (2) clipping per-sample gradients to bound sensitivity, and (3) adding calibrated Gaussian noise (Bagdasaryan et al., 2019; Dwork et al., 2006; Abadi et al., 2016; Bu et al., 2020; Dong et al., 2022; Wang et al., 2019). To track privacy guarantees, advanced

accounting methodologies such as Rényi DP (Mironov, 2017; Wang et al., 2019; Abadi et al., 2016), moments accountant (Koskela et al., 2020; Gopi et al., 2021; Zhu et al., 2022), and Gaussian DP (Dong et al., 2022; Bu et al., 2020) have been developed (Bu et al., 2024).

An important concern is that DP models often underperform compared to their non-private counterparts, particularly in healthcare settings where data tend to be imbalanced and show long-tail distributions (Suriyakumar et al., 2021). Several studies have shown that privacy-preserving mechanisms can negatively affect fairness (Tran et al., 2021), and that different groups may be impacted in unequal ways (Pujol et al., 2020). For example, Bagdasaryan et al. (2019) and Farrand et al. (2020) observed that models trained with DP-SGD experience disproportionate accuracy loss across demographic groups, with underrepresented populations being most affected. One of the main reasons behind this discrepancy is that while gradient clipping is necessary to ensure privacy, it tends to disproportionately suppress the weaker and less frequent gradient signals from underrepresented groups (Esipova et al., 2022; Tran et al., 2021). Therefore, although clipping is vital for privacy, it can lead to fairness degradation (Esipova et al., 2022; Tran et al., 2021). For example, groups with naturally larger gradient norms may experience more aggressive clipping, thereby reducing their representation in model updates and leading to a drop in predictive accuracy for these populations (Bagdasaryan et al., 2019; Xu et al., 2020).

Although this issue has received increasing attention, most existing work focuses on either synthetic benchmarks or general ML datasets. There is a knowledge gap regarding the effects of differentially private training on fairness in real-world scenarios, especially in critical fields like healthcare, where datasets are both structured and unstructured. When ML models perform unevenly across demographic groups, they can produce biased predictions that lead to unfair or inappropriate treatment decisions, ultimately putting people at greater risk of harm (Chouldechova & Roth, 2020; Mehrabi et al., 2021). In this work, we propose **SoftAdaClip**, a novel differentially private training method that integrates a smooth $\tanh$-based transformation into adaptive clipping. We evaluate its fairness and utility across three diverse datasets—the MIMIC-III clinical text dataset (Johnson et al., 2016b;a; Goldberger et al., 2000b), the GOSSIS-1-eICU structured healthcare dataset (Raffa et al., 2022a; Goldberger et al., 2000a), and the Adult Income tabular dataset (Becker & Kohavi, 1996), a standard benchmark in fairness research.

We then investigate whether demographic subgroups (differentiated by attributes such as race, gender, or age) experience uneven performance degradation under DP training. Following the analysis framework of Tran et al. (2021), we test the hypothesis that unequal gradient clipping across subgroups leads to performance differences and fairness issues. Building on prior work in adaptive clipping (Andrew et al., 2021), our method replaces the standard hard clipping function with a smooth $\tanh$-based transformation. This design preserves relative gradient magnitudes while bounding sensitivity for DP, and as a result, helps to avoid suppressing important learning signals from specific subgroups. We demonstrate that SoftAdaClip not only reduces subgroup disparities but also improves overall model utility by decreasing the total loss.

## 2    RELATED WORKS

Although DP has become an essential approach for protecting data confidentiality, in practice, it can introduce a critical trade-off between utility and privacy; in other words, adding noise to gradients can degrade model performance, particularly for underrepresented groups. Bagdasaryan et al. (2019) were among the first to show that differential privacy can disproportionately harm minority groups. They showed that while DP can protect privacy uniformly, its impact on model performance is not evenly distributed across all subgroups. The reason for this unfairness is shown by Esipova et al. (2022); Tran et al. (2021), and it is because minority groups often have larger gradient norms and thus are more aggressively clipped, and this results in diminished representation in model updates. This introduced the concept of "gradient misalignment," where gradients from different groups diverge in direction, and clipping disproportionately suppresses those of minority subpopulations. Building on this, Esipova et al. (2022) provided a detailed analysis of gradient dynamics under DP-SGD, identifying "inequitable clipping" as a primary source of disparate impact. They demonstrated that per-sample clipping can bias the optimization trajectory toward majority group directions, especially when minority gradients are systematically clipped more aggressively. In their study, they analyzed the angle between subgroup gradients and showed that after clipping, the minority

gradient is shrunk more, so the final average is pulled toward the majority direction. They proposed DPSGD-Global-Adapt as a solution, and by measuring directional fairness metrics, they provided a deeper understanding of how DP influences learning dynamics across groups. In healthcare settings, this issue becomes much more critical because of the very high stakes of clinical decision-making and the inherent imbalance in electronic health record (EHR) datasets. Suriyakumar et al. (2021) studied the impact of DP on fairness in clinical data, and showed that privacy mechanisms can affect which patient groups influence model predictions. These results hold for other clinical prediction tasks, thereby suggesting that DP may even increase biases if fairness is not explicitly addressed.

## 2.1 Existing Clipping Methods and Their Shortcomings

Differentially Private SGD (DP-SGD), as introduced by Abadi et al. (2016), clips each per-sample gradient to a fixed L2 norm $C_0$, then averages and perturbs the gradients using Gaussian noise. The clipping threshold, $C_0$, is a fixed hyperparameter that is manually set by the user, and its value is very important because it strongly influences both the convergence speed and the performance of the model after training. If $C_0$ is too small, many gradients are excessively truncated, which can degrade model utility. Conversely, if $C_0$ is too large, most gradients remain unclipped. This situation is problematic because the noise added follows the distribution $\mathcal{N}(0, \sigma^2 C_0^2 I)$; therefore, a larger $C_0$ results in injecting more noise, which can degrade the signal-to-noise ratio (Andrew et al., 2021). Prior work has shown that selecting an appropriate $C_0$ is difficult and task-dependent, especially in settings where gradient norms follow heavy-tailed distributions (Xia et al., 2023). The complete procedure for DP-SGD is summarized in Algorithm 1.

One downside of fixed clipping is that it may intensify fairness issues. When subgroups have different gradient norm distributions, applying a uniform clipping threshold can disproportionately suppress updates for one group (e.g., minorities), which can hurt the utility for that group. Bagdasaryan et al. (2019) were among the first to show that DP-SGD worsens accuracy disparities in such settings. To address these limitations, Andrew et al. (2021) proposed an adaptive clipping strategy that adjusts the threshold over time based on the distribution of gradient norms using a differentially private estimate of a target quantile. This adaptive strategy, as shown in Algorithm 2, reduces the need for manual tuning and aligns the clipping norm more closely with the actual distribution of gradients, and therefore improves the trade-off between privacy and

---

**Algorithm 1** DPSGD

**Require:** Iterations $T$, Dataset $D$, sampling rate $q$, clipping bound $C_0$, noise multiplier $\sigma$, learning rates $\eta_t$
**Initialize:** $\theta_0$ randomly
1: **for** $t = 0, \ldots, T - 1$ **do**
2:     $B \leftarrow$ Poisson sample of $D$ with rate $q$
3:     **for** each $(x_i, y_i) \in B$ **do**
4:         $g_i \leftarrow \nabla_\theta \ell(f_\theta(x_i), y_i)$
5:         $\bar{g}_i \leftarrow g_i \cdot \min\left(1, \frac{C_0}{\|g_i\|}\right)$
6:     **end for**
7:     $\tilde{g}_B \leftarrow \frac{1}{|B|}\left(\sum_{i \in B} \bar{g}_i + \mathcal{N}(0, \sigma^2 C_0^2 I)\right)$
8:     $\theta_{t+1} \leftarrow \theta_t - \eta_t \tilde{g}_B$
9: **end for**

---

utility (Xia et al., 2023). Andrew et al. (2021) showed that clipping to a data-driven quantile can outperform even the best fixed clip chosen in hindsight.

While adaptive clipping improves utility and tuning flexibility, it still applies a uniform hard threshold to all gradients, which raises fairness concerns. Specifically, gradients with norms exceeding the dynamic threshold $C$ are all scaled to have the same norm, regardless of their original size or informativeness. This uniform compression can disproportionately affect underrepresented groups, whose gradients are often larger in magnitude due to harder-to-learn patterns or lower representation in the data. As a result, their distinct learning signals may be suppressed. For example, if $C$ is smaller than both 20 and 1000, then two gradients with norms 20 and 1000 would both be clipped to $C$, despite one potentially carrying a stronger signal. In contrast, gradients from majority groups that fall below $C$ remain unaltered, preserving their expressiveness. This imbalance may unintentionally bias learning toward majority patterns, thereby perpetuating inequalities and undermining fairness.

## 3 Improving fairness with Soft Clipping

To address the fairness limitations of hard thresholding, we propose **SoftAdaClip**, which replaces sharp clipping with a smooth, nonlinear transformation using the hyperbolic tangent ($\tanh$) function.

Our method aims to preserve the relative magnitudes of large gradients while still bounding their norms for differential privacy.

Let $g_i$ denote the gradient for the $i^{\text{th}}$ sample, and let $\|g_i\|_2$ be its $\ell_2$-norm. This norm is used for clipping as described in Algorithm 2.

We then define a scaling factor based on the $\tanh$ function:

$$\alpha_i = \tanh\left(\frac{C}{\|g_i\|_2 + \epsilon}\right)$$

Where $C$ is the target sensitivity (similar to the clipping bound in DP-SGD) and $\epsilon$ is a small constant (e.g., $10^{-6}$) to avoid division by zero.

The final transformed gradient is:

$$\bar{g}_i = \alpha_i \cdot g_i = \tanh\left(\frac{C}{\|g_i\|_2 + \epsilon}\right) \cdot g_i$$

This transformation adapts based on the gradient norm. For small gradients where $\|g_i\|_2 \ll C$, we have $\alpha_i \approx 1$, so the gradient is minimally affected. For large gradients where $\|g_i\|_2 > C$, the $\tanh$ function smoothly compresses the norm instead of mapping all high-norm gradients to the same clipped value.

Despite this change, the method still preserves privacy because the per-sample sensitivity remains bounded by $C$. Since $\tanh(x) < \min(x, 1)$ for $x > 0$, it follows that:

$$\|\bar{g}_i\|_2 = \alpha_i \|g_i\|_2 = \tanh\left(\frac{C}{\|g_i\|_2 + \epsilon}\right) \|g_i\|_2 \leq C.$$

**Algorithm 2** ADAPTIVE (ANDREW ET AL.) VS. SOFTADACLIP

**Require:** Iterations $T$, Dataset $D$, sampling rate $q$, initial clipping bound $C$, noise multipliers $\sigma$, $\sigma_b$, learning rates $\eta_t$, $\eta_C$, target quantile $\gamma$

**Initialize:** $\theta_0$ randomly, set clipping bound $C \leftarrow C_0$

1: **for** $t = 0, \ldots, T-1$ **do**
2: $\quad B \leftarrow$ Poisson sample of $D$ with rate $q$
3: $\quad$ **for** each $(x_i, y_i) \in B$ **do**
4: $\quad\quad g_i \leftarrow \nabla_\theta \ell(f_\theta(x_i), y_i)$
5: $\quad\quad$ # Original hard clipping
6: $\quad\quad \bar{g}_i \leftarrow g_i \cdot \min\left(1, \frac{C}{\|g_i\|}\right)$
7: $\quad\quad$ # smooth clipping via tanh
8: $\quad\quad \alpha_i \leftarrow \tanh\left(\frac{C}{\|g_i\| + \varepsilon}\right)$
9: $\quad\quad \bar{g}_i \leftarrow \alpha_i \cdot g_i$
10: $\quad\quad b_i \leftarrow \mathbf{1}\{\|g_i\| \leq C\}$
11: $\quad$ **end for**
12: $\quad \tilde{g}_B \leftarrow \frac{1}{|B|}\left(\sum_{i \in B} \bar{g}_i + \mathcal{N}(0, \sigma^2 C^2 I)\right)$
13: $\quad \theta_{t+1} \leftarrow \theta_t - \eta_t \tilde{g}_B$
$\quad\quad\quad \triangleright$ Adaptively update clipping bound $C$ to match target quantile $\gamma$
14: $\quad \tilde{b}_t \leftarrow \frac{1}{|B|}\left(\sum b_i + \mathcal{N}(0, \sigma_b^2)\right)$
15: $\quad C \leftarrow C \cdot \exp(-\eta_C \cdot (\tilde{b}_t - \gamma))$
16: **end for**

which ensures the clipping bound is never exceeded. In this way, SoftAdaClip retains the privacy guarantees of DP-SGD while avoiding the hard cutoff. As shown in Figure 1, hard clipping flattens all values beyond the threshold, whereas SoftAdaClip grows smoothly and maintains distinct outputs.

For example, consider $\|g_i\|_2 = 1.1$ and $\|g_i\|_2 = 1.2$ with $C = 1$. hard clipping maps both to $\bar{g}_i$ with norm 1, whereas SoftAdaClip assigns scaling factors of $\tanh(0.91) \approx 0.72$ and $\tanh(0.83) \approx 0.68$, thereby preserving their distinction. Therefore, this approach maintains bounded sensitivity while preserving intra-group variability and enabling stronger learning signals from minority subpopulations.

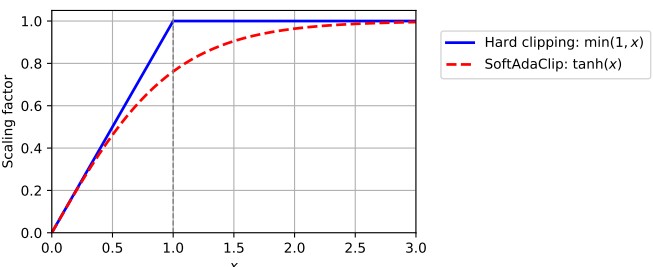

Figure 1: Comparison of hard clipping ($\min(1, x)$) and SoftAdaClip ($\tanh(x)$). Hard clipping maps all gradients above the threshold to the same value, while SoftAdaClip smoothly compresses them, preserving differences.

As in Andrew et al. (2021), we adaptively update the clipping threshold $C$ using a differentially private estimate of the proportion of large gradients that exceed a specified tolerance threshold. The update rule is defined as follows:

$$C \leftarrow C \cdot \exp(-\eta_C \cdot (\hat{b}_t - \gamma))$$

In this equation, $\hat{b}_t$ is a differentially private estimate of the fraction of gradients exceeding the current threshold, $\gamma$ is the target unclipped ratio (e.g., median), and $\eta_C$ is a learning rate for the threshold. Unlike the adaptive clipping algorithm proposed by Andrew et al. (2021), our approach does not reduce all large gradients to a uniform norm. Instead, it maintains relative differences through a smooth tanh-based scaling function. This approach enables more expressive updates, particularly for underrepresented groups whose gradients tend to be suppressed under hard clipping. As a result, SoftAdaClip improves both utility and fairness. The complete procedure is presented in Algorithm 2, with our $\tanh$-based modifications highlighted in red.

## 4 EXPERIMENTAL SETUP

To evaluate the effectiveness of our proposed SoftAdaClip method, we compare it against both standard DP-SGD and Adaptive-DPSGD (Andrew et al., 2021). We conduct experiments on three diverse datasets, including MIMIC-III (for length of stay (LOS) prediction), eICU (for ICU mortality prediction), and Adult Income (for income classification), to cover a range of domains and data modalities. All experiments are conducted with and without differential privacy, and private models are trained with five different seeds, over which we report the mean performance. Full preprocessing details for all datasets are provided in Appendix A.

### 4.1 MIMIC-III (LENGTH OF STAY PREDICTION)

MIMIC-III contains clinical notes and structured ICU information for over 40,000 patients (Johnson et al., 2016b;a; Goldberger et al., 2000b). We train a LOS prediction task using this dataset, and we preprocess it by following the approach introduced in the Clinical Outcome Prediction benchmark by van Aken et al. (2021). Specifically, we follow their method of constructing an adult cohort with valid discharge summaries and preparing the LOS labels in four ordinal classes. We also retain demographic attributes for downstream fairness analysis. For model training, we fine-tune the ClinicalBERT model. During training, the transformer layers are kept frozen, and only the classification head is updated. The non-private model is trained for 20 epochs with a batch size of 40 and a learning rate of 1e-5. In the differentially private setting, due to performance degradation caused by noise injection, we increase the batch size to 500, the learning rate to 1e-3, and extend training to 200 epochs. These adjustments are standard, since noise can confuse the optimization process; therefore, a higher learning rate with a bigger batch size and longer training duration help maintain performance.

### 4.2 EICU (MORTALITY PREDICTION)

The GOSSIS-1-eICU dataset (Raffa et al., 2022a;b) is a model-ready, preprocessed subset of the eICU database (Pollard et al., 2018), covering ICU stays from 204 U.S. hospitals. We follow the released model-ready version and apply additional filtering, age binarization, one-hot encoding, and balancing steps to create a stable classification setting for differential privacy. For modelling, we implement a fully connected neural network with four hidden layers (128, 64, 32, 2), with ReLU activations, GroupNorm in the first two layers, and dropout (0.3) before the final classifier. The output layer produces two logits, and training uses CrossEntropyLoss with class weights [0.5, 1.0]. The non-private model is trained for 30 epochs with a batch size of 32 and a learning rate of 1e-5. The private model uses a batch size of 64, learning rate of 1e-3, and 60 epochs.

### 4.3 ADULT INCOME (INCOME CLASSIFICATION)

The Adult Income dataset (Becker & Kohavi, 1996), includes demographic and employment information used to predict whether an individual earns more than $50K per year. We use the Adult

Income dataset, preprocessed following Esipova et al. (2022) and Le Quy et al. (2022). We evaluate this task using two models. The simple model is a two-layer neural network with 256 hidden units per layer and ReLU activation, followed by a sigmoid output, trained using BCEWithLogitsLoss with a positive-class weight of 2.0. This architecture matches the simple model used by Esipova et al. (2022). The complex model mirrors the architecture used for the eICU dataset and includes GroupNorm layers, ReLU activations, and a final softmax output, trained using CrossEntropyLoss with class weights [1.0, 2.0]. The simple model is trained for 15 epochs (batch size = 32, learning rate = 0.0003), and its private counterpart is trained for 30 epochs (batch size = 128, learning rate = 0.003). The non-private complex model is trained for eight epochs (batch size = 32, learning rate = 3e-5), and the private version for 16 epochs (batch size = 128, learning rate = 0.0003).

For all datasets and model variants, hyperparameters for both private and non-private models were selected through extensive hyperparameter tuning using Optuna to ensure stable and comparable performance across settings.

## 4.4 Differential Privacy Settings

All private models used DP-Adam and were trained with the Opacus differential privacy library (Yousefpour et al., 2021) with $\epsilon = 8$, max_grad_norm = 0.1, and $\delta$ fixed at $1 \times 10^{-5}$, which is on the order of $1/n$ and follows standard practice (Esipova et al., 2022). As shown in prior work, smaller $\epsilon$ values tend to improve fairness (Tran et al., 2025) but reduce overall utility, whereas larger $\epsilon$ values provide stronger utility at the expense of privacy. The noise multiplier is computed jointly from $\epsilon$, the sampling rate, and the clipping threshold using the RDP accountant, and its scale increases proportionally with the clipping norm. Following the literature, we adopt $\epsilon = 8$ as a practical midpoint that offers a reasonable trade-off between privacy, fairness, and utility, which is why we use this value in all of our experiments. Based on findings in prior work, we use $C_0 = 0.1$ as a strong baseline and later explore smaller thresholds to assess their effect in our context. Our choice of clipping threshold, optimizer, and other DP hyperparameters follows prior findings from Li et al. (2021); Bu et al. (2024), and related work demonstrating that smaller clipping thresholds and properly tuned DP-Adam improve utility under differential privacy. We adopt these settings accordingly, and the full reasoning and supporting evidence from the literature are provided in Appendix B. To enable large batch sizes, we used Opacus's BatchMemoryManager. Early stopping based on validation F1-score was applied in different experiments to prevent overfitting, and hyperparameters were tuned with Optuna (Akiba et al., 2019).

## 4.5 Gradient Behaviour Analysis

Following the approach of Esipova et al. (2022), we conduct a detailed analysis of gradient norms to investigate whether groups experiencing greater gradient clipping also suffer from larger performance degradation. To isolate and evaluate the behaviour of the trained models on different subgroups, we define subgroups based on sensitive attributes, including gender, age, and ethnicity. We measure the L2 norms of gradients before and after clipping in the differentially private training pipeline for each subgroup. Specifically, we calculate the overall gradient norm before clipping and noise addition by accumulating the per-parameter gradients across microbatches, flattening and concatenating them, and calculating their L2 norm. After applying the clipping (but before adding noise), we again calculate the norm of the resulting gradients. By comparing these two values, we assess how much of the original gradient magnitude was reduced by the clipping mechanism for each subgroup.

Our analysis, which aligns with the findings of Esipova et al. (2022), also reveals that subgroups whose gradients are clipped more aggressively tend to have higher loss values. This suggests that these subgroups originally had higher gradient magnitudes, which were suppressed by the clipping operation, leading to degraded performance. We use the proposed SoftAdaClip to address this problem. To quantify the amount of clipping applied, we compute the absolute difference between the gradient norms before and after clipping for each subgroup and compare it with other approaches. The results show that SoftAdaClip consistently applies less clipping than other approaches. This is due to its smooth, non-binary transformation, which better preserves gradient direction and maintains learning signals for subgroups with high gradients, ultimately improving fairness. The full table reporting gradient norms before and after clipping for each dataset and subgroup is provided in Appendix C.

# 5 RESULTS

Our evaluation focuses on two primary metrics: (1) overall loss, and (2) loss gap, defined as the absolute difference in loss between subgroups (e.g., male vs. female). In our experiments, we compute loss using the summation form of the loss functions, defined as:

$$\mathcal{L} = \sum_{i=1}^{N} \ell(f_\theta(x_i), y_i)$$

Where $\ell(f_\theta(x_i), y_i)$ denotes the per-sample loss between the model's prediction $f_\theta(x_i)$ and the actual label $y_i$, and $N$ is the total number of examples in the evaluation set.

Figure 2 presents the loss gap across different subgroups (gender, age, ethnicity) for three datasets (eICU, Income, and MIMIC), using three clipping strategies: standard DPSGD, Andrew et al. (2021) Adaptive-DPSGD, and our proposed method SoftAdaClip. For the Income dataset, we report results for both the simple and complex model variants. All results are averaged over five different random seeds to assess generalizability. In 7 of 9 cases, SoftAdaClip achieves the lowest loss gap.

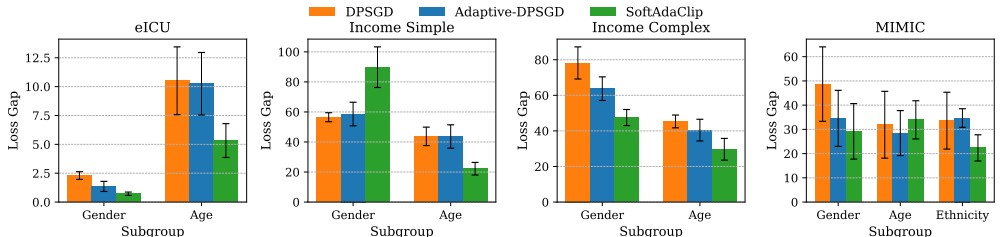

Figure 2: Comparison of loss gaps between subgroups across different datasets and clipping strategies. The figure presents the subgroup loss disparities for three differentially private training methods: DPSGD, Andrew et al. (2021) Adaptive-DPSGD, and SoftAdaClip.. Error bars represent ±1 standard error of the mean, computed across five random seeds.

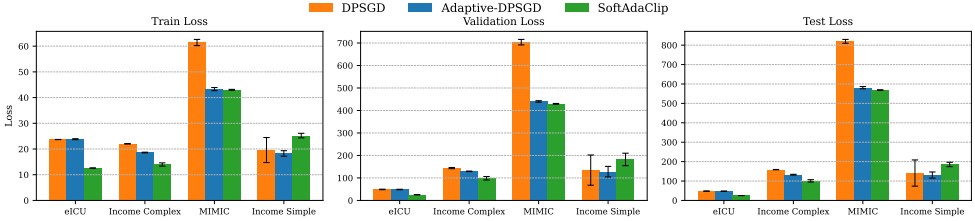

Figure 3: Comparison of train, validation, and test losses across datasets using DPSGD, Andrew et al. (2021) Adaptive-DPSGD, and SoftAdaClip. Results are averaged over five random seeds. Error bars represent ±1 standard error of the mean.

Figure 3 compares the train, validation, and test losses for all datasets and clipping strategies, averaged over five seeds. SoftAdaClip consistently shows lower losses across the train, validation, and test sets, except for the Income Simple model. We calculated the gradient norms during training and noticed that the income dataset started with a significantly lower gradient norm compared to other datasets.

This led us to believe that the increase in loss and the lack of improvement with softAdaClip in this setting are due to its lower gradient norms. Since SoftAdaClip applies a $\tanh$-based transformation, gradients that are smaller than the clipping threshold $C_0$ are slightly reduced, which may reduce useful signal in this scenario with small gradient norms. To test this hypothesis, we re-ran the Income Simple model with smaller clipping thresholds ($C_0 = 0.01$ and $0.05$). As shown in Figure 4, the results confirm our expectation. We observe that the overall loss for testing, training, and validation phases with SoftAdaClip decreases compared to the original threshold setting. Additionally, Figure 5 shows that the loss gap for different subgroups with SoftAdaClip also decreases compared to the original (larger) threshold setting. While the reduction in overall loss is modest, it stands in contrast to earlier results, where higher clipping thresholds led to an increase in loss. These findings demonstrate

that adjusting the clipping threshold can mitigate the unintended reduction of already small gradients and improve both utility and fairness in low-gradient settings.

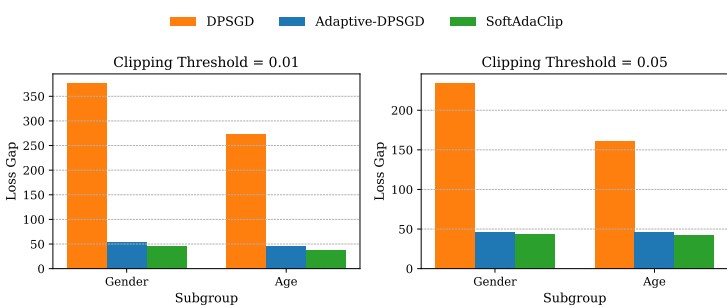

Figure 4: Comparison of loss gaps between subgroups in the Income Simple model using clipping thresholds of 0.01 and 0.05 across three differentially private training methods: DPSGD, Andrew et al. (2021) Adaptive-DPSGD, and SoftAdaClip.

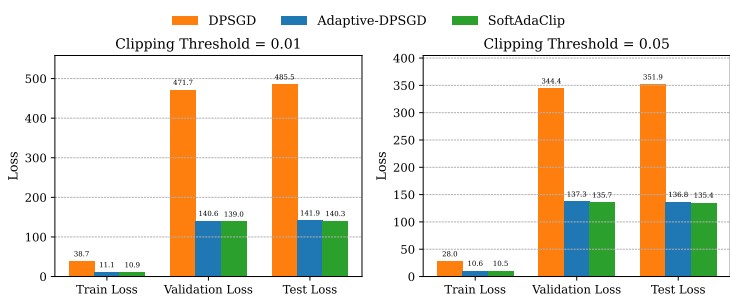

Figure 5: Comparison of train, validation, and test losses in the Income Simple model using clipping thresholds of 0.01 and 0.05 across three differentially private training methods: DPSGD, Andrew et al. (2021) Adaptive-DPSGD, and SoftAdaClip.

Moreover, we also calculated the percentage improvements in average subgroup disparity. SoftAdaClip reduces disparities by up to 87% compared to DPSGD and up to 48% compared to Adaptive-DPSGD, demonstrating consistent fairness gains across most datasets. The full formulas, tables, and dataset-specific results are provided in Appendix C.

### 5.1 STATISTICAL SIGNIFICANCE OF DISPARITY REDUCTIONS

To evaluate whether there are any statistically significant differences between losses across subgroups, we use pairwise Wilcoxon signed-rank tests. We compare the loss gaps across all datasets and demographic subgroups for different methods, including Adaptive Clipping, SoftAdaClip, and DP-SGD. We choose the Wilcoxon test as a non-parametric alternative to the paired $t$-test because it does not depend on a normal distribution, which would not applicable for our data. Also, it is designed explicitly for paired comparisons, which aligns perfectly with our setup. In our analysis, we directly compare each subgroup result from one method to the corresponding result from another method, across the same seed, dataset, and subgroup. To address the issue of multiple comparisons, we apply Bonferroni correction. The results of our analysis indicates that SoftAdaClip significantly outperforms both Adaptive Clipping and DP-SGD in reducing subgroup loss gaps, with $p$-values of less than 0.001 and 0.0004, respectively. In contrast, there is no statistically significant difference between Adaptive Clipping and DP-SGD, with a $p$-value of 1.0. These findings suggest that SoftAdaClip is more effective at mitigating disparities across subgroups compared to the baseline methods.

### 5.2 COMPARISON WITH PRIOR WORK

We compare our findings with the results reported by Esipova et al. (2022) for the DPSGD-Global-Adapt method on the Adult Income dataset. for this comparison, we adjust to reduction=`mean` by

dividing by the batch size, because Esipova et al. (2022) report mean losses rather than summed losses, and matching their reduction setting is necessary to ensure a fair, like-for-like comparison. According to their paper, the female loss was 0.18, the male loss was 0.39, and the resulting subgroup disparity (loss gap) was 0.21. In comparison, our SoftAdaClip method with a simple approach and a lower clipping threshold achieves a female loss of 0.364 and a male loss of 0.532, resulting in a loss gap of 0.168, which is less disparity than Esipova et al. (2022) reports. Based on these results, SoftAdaClip seems to be more effective at reducing subgroup disparities and improving fairness, and this further validates the advantage of combining adaptive clipping with smooth soft clipping. We suggest that future research explore additional comparison with the method proposed by Esipova et al. (2022), and whether integrating their approach with soft (tanh-based) clipping could lead to further improvements in fairness.

A broader comparison can be made with Pannekoek & Spigler (2021), who evaluated fairness on Adult Income using risk difference (positive prediction rate gaps) rather than loss-based disparity. Their S-NN (simple neural network) achieved 84.14% accuracy, while their DP-NN variant reached 84.03%. Fairness-constrained models, including Reject Option Classification (Fair-NN) and DPF-NN (DP and Fair), incurred larger accuracy declines, with Fair-NN dropping to 79.25% and DPF-NN to 82.98%. Notably, DPF-NN reduced risk difference below 0.05 while incurring only a 1.16 percentage-point accuracy decrease (84.14 → 82.98). Our results exhibit a similar pattern: in the Simple Income setting, accuracy declines by less than 1 percentage point (0.8544 → 0.8447), and in IncomeComplex, by approximately 1.4 points (0.8501 → 0.8362). This suggests that our simpler models experience even smaller utility degradation than DPF-NN relative to their own baselines. Consistent with their observation that simpler models often improve fairness, our simple configuration delivers strong loss-gap reductions while maintaining accuracy highly comparable to non-private training.

Another important comparison is with Jagielski et al. (2019), who study the privacy–fairness–accuracy trilemma and show that DP can substantially worsen accuracy and fairness on smaller datasets. They propose DP post-processing and a DP oracle-learner to enforce equalized odds, but these approaches require access to sensitive attributes at test time or suffer from instability due to noise. In contrast, SoftAdaClip modifies the core DP-SGD mechanism itself, specifically, the clipping operation, to reduce subgroup distortions without requiring post-hoc adjustments or test-time attributes. Empirically, while Jagielski et al. (2019) highlight that DP often harms accuracy on small datasets, our results show that with large-scale datasets like MIMIC-III, eICU, and Adult Income, smoother adaptive clipping improves fairness (up to 87% reduction in subgroup loss disparity on Adult Income) while keeping accuracy within 1–2 percentage points of non-private baselines.

Finally, recent work such as FAIRDP (Tran et al., 2025) shows that stricter privacy budgets (smaller $\epsilon$) can improve demographic-parity fairness by up to 75% with only a 2–5% utility reduction. Although their evaluation focuses on prediction-rate fairness metrics, whereas we study loss-based subgroup disparities, the high-level insights are complementary: both results indicate that well-designed DP mechanisms can simultaneously improve fairness and preserve utility. SoftAdaClip contributes to this growing body of evidence by showing that smoothing the gradient clipping step yields substantial improvements in loss-parity fairness while maintaining accuracy within a narrow margin of non-private models.

Overall, these comparisons demonstrate that SoftAdaClip achieves fairness improvements competitive with, and in several cases exceeding, state-of-the-art DP-fairness methods, while maintaining accuracy within 1–2 percentage points of non-private baselines across multiple datasets and model classes.

## 5.3 Ablation Study: Effect of Smoothing Without Adaptivity

To better understand the source of fairness improvements, we conducted an ablation study to disentangle the effect of the tanh-based smoothing from the adaptive thresholding mechanism. Specifically, we evaluated a variant of DPSGD that uses fixed clipping thresholds combined with tanh-based scaling. With this new approach, we isolated the effect of smooth clipping. Results show that smoothing alone does not consistently reduce subgroup disparities across datasets, whereas SoftAdaClip, which combines smoothing with adaptive thresholding, achieves consistent improvements. This ablation highlights that adaptivity is a critical factor in the design, and the tanh transformation by itself is insufficient to improve fairness. Full details and results of this evaluation, including the complete set of experimental results and corresponding tables, are provided in Appendix C.

## 6    LIMITATIONS

The present findings highlight the promise of SoftAdaClip, but also point to important directions for extending and validating the method. First, our evaluation was limited to three datasets (MIMIC-III, eICU, and Adult Income), covering both healthcare text and tabular data. Although the $\mathtt{tanh}$-based transformation we propose is not specifically linked to a single data type, and its effects on gradients should, in principle, remain consistent across different modalities, our experiments did not include vision or multi-modal datasets. It would still be beneficial to explore the application of SoftAdaClip in these contexts to evaluate its generalizability and to ensure that the improvements in fairness and utility extend beyond the domains we have examined here.

Our implementation is grounded in the adaptive clipping method introduced by Andrew et al. (2021). While this serves as a solid baseline, we did not examine the impact of $\mathtt{tanh}$ smoothing when used alongside other adaptive or privacy-preserving techniques. It is possible that combining SoftAdaClip with other strategies could achieve even stronger improvements in fairness and utility.

Third, we found that in low-gradient scenarios, like the Income Simple model, achieving fairness improvements necessitated the use of smaller clipping thresholds. Although this adjustment did not adversely affect accuracy in our experiments, it does increase sensitivity to hyperparameter tuning. This heightened sensitivity may be undesirable for users who seek robustness without the need for extensive parameter optimization.

Finally, in certain contexts, SoftAdaClip resulted in a slight decrease in accuracy compared to Adaptive-DPSGD. Throughout all datasets, this reduction never exceeded 1% (Table 1). Although such a reduction is minor, it highlights that fairness gains may sometimes come with a very small utility trade-off. In practice, this trade-off is unlikely to be critical, but it is important to acknowledge.

## 7    CONCLUSION

In this work, we investigated the fairness implications of differentially private training, with a particular focus on the role of gradient clipping. We showed that traditional DP-SGD and existing adaptive clipping methods can exacerbate performance disparities across demographic subgroups due to uneven gradient suppression. To address this, we introduced SoftAdaClip, a novel training approach that combines adaptive clipping with a smooth $\mathtt{tanh}$-based transformation to preserve relative gradient magnitudes while bounding sensitivity for privacy. Empirically, SoftAdaClip consistently reduced subgroup disparities compared to both DPSGD and Adaptive-DPSGD while maintaining rigorous $(\epsilon, \delta)$-differential privacy guarantees. Through extensive experiments on three diverse datasets, we demonstrated that SoftAdaClip reduces subgroup loss disparities and improves overall utility. In terms of fairness, these improvements are important because they show that disparities across different demographic groups can be substantially reduced, demonstrating that differential privacy does not necessarily worsen equity, and that such harms can be mitigated with the right clipping strategy. In terms of privacy, the results show that privacy-preserving training can achieve strong fairness protections without relaxing formal guarantees. Taken together, these findings highlight that privacy-preserving training can deliver fairness gains that were previously thought incompatible with rigorous $(\epsilon, \delta)$-differential privacy guarantees. Ablation studies revealed that smoothing alone, without adaptive thresholding, is insufficient to improve fairness, underscoring the importance of the adaptive component in our design.

Our findings highlight that the choice of clipping strategy is central to balancing privacy, utility, and fairness, and that SoftAdaClip preserves strong differential privacy guarantees while narrowing subgroup loss gaps. As a direction for future research, we encourage exploring the integration of $\mathtt{tanh}$ smoothing with other adaptive clipping strategies to understand its generalizability and impact on fairness. Additionally, evaluating these methods across a broader range of model architectures and domains beyond the datasets examined here will further clarify their practical applicability and limitations.

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

## A APPENDIX: DATASET DETAILS AND PREPROCESSING

This appendix provides additional details on the datasets used in our experiments, as well as the preprocessing pipelines applied to prepare them for differentially private training. Our goal was to ensure consistency across datasets while preserving the features necessary for downstream utility and fairness evaluation. Each dataset required task-specific and modality-specific preprocessing, as outlined below.

## A.1 MIMIC III

The Medical Information Mart for Intensive Care III (MIMIC-III) is a large, publicly available clinical database containing de-identified data for over 40,000 ICU patients admitted to the Beth Israel Deaconess Medical Center between 2001 and 2012 (Johnson et al., 2016b;a; Goldberger et al., 2000b). The dataset includes detailed records such as demographics, vitals, lab results, medications, and free-text clinical notes. For this study, we specifically utilize the NOTEEVENTS table, which contains unstructured notes written by physicians, nurses, and other care providers during patient admissions.

For the LOS prediction task, we primarily followed the preprocessing pipeline introduced in the Clinical Outcome Prediction benchmark by van Aken et al. (2021) as mentioned earlier. In line with their approach, we excluded newborn admissions by removing records with ADMISSION_TYPE = "NEWBORN", and retained only discharge summaries with non-empty TEXT and HADM_ID fields. We removed duplicated notes by keeping only the latest discharge summary per admission and merged multiple notes (e.g., addenda) into a single document. To prevent label leakage, we removed sections that could contain information about discharge outcomes, such as "Discharge Diagnosis," "Discharge Medications," or any section headers containing keywords like "home" or "discharge". We computed the length of stay in days using the ADMITTIME and DISCHTIME fields and excluded hospitalizations with mortality to ensure consistency in the prediction target. Although we did not explicitly filter patients below age 18 (Bao et al., 2023; Pang et al.), or below age 15 Auslander et al. (2020); Meng et al. (2022), the exclusion of newborns serves a similar purpose, aligning with practices in other studies that restrict cohorts to adult populations. Following the benchmark setup, we discretized LOS into four ordinal classes: $\leq 3$ days, 4–7 days, 8–14 days, and >14 days. Additional demographic features (e.g., age, gender, ethnicity, insurance, religion, marital status) were retained from structured data and preprocessed for downstream fairness analysis.

## A.2 GOSSIS-1-EICU

GOSSIS-1-eICU dataset (Raffa et al., 2022a;b), is a preprocessed subset of the eICU Collaborative Research Database (Pollard et al., 2018), which is made available via PhysioNet (Goldberger et al., 2000a). This dataset includes ICU admissions from 2014–2015 across 204 U.S. hospitals, and it is limited to patients over 16 years old with ICU stays longer than six hours. This dataset has complete information on vital signs and outcomes. The model-ready version (gossis-1-eicu-only-model-ready.csv.gz) has already undergone preprocessing, imputation, and feature selection using the rGOSSIS1 package. We further filter the model-ready dataset by removing conflicting labels (e.g., icu_death = 1 and hospital_death = 0) and dropping any remaining rows with missing values. Categorical features, such as diagnosis category, ICU admission source, and clinical severity group, are one-hot encoded. We merge gender, ethnicity, and age from the original raw file, and binarize age using the median (65 years) to create balanced younger and older groups. To address class imbalance for in-hospital mortality prediction, we apply random undersampling to equalize the number of positive and negative samples, resulting in 11,817 deaths and 11,817 survivors. The final dataset includes 12,672 males and 10,948 females, and 12,312 older versus 11,322 younger patients. We split the data into training (70%), validation (10%), and test (20%) sets using stratified sampling. For fairness evaluation, we retain gender and age group as protected attributes but exclude them from model input during training.

## A.3 ADULT INCOME

The Adult Income dataset (Becker & Kohavi, 1996), extracted from the 1994 U.S. Census Bureau database, includes demographic and employment data to predict whether an individual earns over $50K per year. It contains cleaned records filtered by age, income, and work hours, and is widely used in fairness research due to its sensitive attributes like sex, race, and age. We use the Adult Income dataset, preprocessed following Esipova et al. (2022) and Le Quy et al. (2022). After removing missing values and duplicates, we reduce the dataset to 45,222 samples. Categorical features are consolidated (e.g., education, marital status, race, work class), numerical variables are normalized or binarized, and one-hot encoding is applied where needed. Outliers such as extreme capital-gain values are excluded. The hours-per-week feature is grouped into categorical bins following (Le Quy et al., 2022), while the age variable is binarized based on the median to create a balanced representation of

age groups. The task is binary income classification ($\leq \$50K$ vs. $> \$50K$). For fairness evaluation, we use sex and age as protected attributes. To control for class imbalance, we follow Esipova et al. (2022) and subsample a balanced dataset with approximately 14,000 male and 14,000 female samples.

The eICU mortality labels were balanced because the original distribution is highly imbalanced and leads to instability under differential privacy. In contrast, the label imbalance in MIMIC-III and Adult Income is considerably milder, and prior work on these tasks consistently trains models using their natural distributions. To maintain comparability with the literature and because the natural imbalance does not hinder model performance in these settings, we retained the original label frequencies for these datasets.

## B BACKGROUND ON DIFFERENTIAL PRIVACY TRAINING AND HYPERPARAMETER CHOICES

This section contains the theoretical motivation and prior work supporting our DP hyperparameter choices. As mentioned earlier, models trained with Differential Privacy (DP) often experience a drop in performance. However, Li et al. (2021) showed that this drop can be mitigated through: (1) using large pretrained language models, (2) tuning hyperparameters specifically for DP, and (3) adopting fine-tuning objectives that closely align with the model's pretraining process. Although these strategies help recover utility, the computational cost of DP training remains significantly higher than that of non-private training, especially for large-scale models (Mehta et al., 2022). A key hyperparameter in this context is the clipping threshold $C_0$, which controls how privacy affects learning. Tuning $C_0$ is a labour-intensive and costly process, but it strongly influences model performance (Bu et al., 2024). Recent studies have found that state-of-the-art (SotA) DP performance is often achieved with smaller clipping thresholds. For example, Li et al. (2021) reported optimal results using $C_0 = 0.1$ with GPT-2 and RoBERTa on NLP tasks, and similar findings were reported in vision tasks such as CIFAR-10 using ResNeXt-29 and SimCLRv2 (Chen et al., 2020; Bu et al., 2024). Based on these findings, we use $C_0 = 0.1$ as a strong baseline and later explore smaller thresholds to assess their effect in our context. Additionally, Li et al. (2021) found that properly tuned DP-Adam can significantly improve performance and even approach non-private baselines. Motivated by this, we use DP-Adam as our optimizer in all differentially private models.

## C FULL EXPERIMENTAL RESULTS

Table 1 presents the test, train, and validation loss values across all datasets and training methods, including Non-Private, DPSGD, Adaptive Clipping, and our proposed SoftAdaClip. Across nearly all settings, SoftAdaClip achieves significantly lower losses compared to prior private methods, demonstrating improved generalization.

An exception occurs with the Income Simple dataset at a higher clipping threshold ($C = 0.1$), where SoftAdaClip does not outperform other private methods. As discussed in the main paper, this is due to the relatively small gradient norms in this dataset, which makes a large clipping threshold suboptimal. In such cases, selecting a smaller clipping bound can lead to better privacy-utility trade-offs. This highlights the importance of tuning the clipping threshold appropriately for the dataset characteristics.

Table 1: Accuracy, F1-score, and loss metrics across datasets and methods.

| Dataset | Method | Accuracy | F1-score | Test Loss | Train Loss | Val Loss |
|---------|--------|----------|----------|-----------|------------|----------|
| eICU ($C = 0.1$) | Non-Private | 0.7883 | 0.8074 | 9.5053 | 9.3519 | 9.9008 |
| | DPSGD | 0.7838 | 0.7857 | 48.0517 | 23.6906 | 48.9455 |
| | Adaptive | 0.7848 | 0.7856 | 47.4854 | 23.8187 | 48.6887 |
| | SoftAdaClip | 0.7786 | 0.7835 | 25.1988 | 12.5813 | 25.5969 |
| Income Simple ($C = 0.1$) | Non-Private | 0.8544 | 0.8251 | 13.8389 | 13.5270 | 14.5337 |
| | DPSGD | 0.8493 | 0.8032 | 141.1011 | 19.6013 | 134.8455 |
| | Adaptive | 0.8518 | 0.8127 | 130.8347 | 18.2785 | 128.0605 |
| | SoftAdaClip | 0.8492 | 0.7975 | 185.3278 | 25.2212 | 182.2780 |
| Income Simple ($C = 0.01$) | DPSGD | 0.6282 | 0.0129 | 485.4643 | 38.6855 | 471.7071 |
| | Adaptive | 0.8471 | 0.8065 | 141.8682 | 11.0803 | 140.6238 |
| | SoftAdaClip | 0.8447 | 0.8057 | 140.2801 | 10.9312 | 138.9555 |
| Income Simple ($C = 0.05$) | DPSGD | 0.6854 | 0.3081 | 351.8910 | 27.9915 | 344.4491 |
| | Adaptive | 0.8439 | 0.8047 | 136.7563 | 10.6433 | 137.3403 |
| | SoftAdaClip | 0.8438 | 0.8055 | 135.4402 | 10.5325 | 135.6755 |
| Income Complex ($C = 0.1$) | Non-Private | 0.8501 | 0.8213 | 14.0957 | 13.7800 | 14.6519 |
| | DPSGD | 0.8358 | 0.7871 | 158.8316 | 22.0290 | 143.8559 |
| | Adaptive | 0.8294 | 0.7797 | 132.7497 | 18.6308 | 129.3976 |
| | SoftAdaClip | 0.8362 | 0.7863 | 100.9126 | 13.9779 | 98.7079 |
| MIMIC ($C = 0.1$) | Non-Private | 0.8720 | 0.8669 | 12.6840 | 14.8288 | 13.0512 |
| | DPSGD | 0.6695 | 0.6034 | 819.7089 | 61.4040 | 703.7503 |
| | Adaptive | 0.7419 | 0.7148 | 580.9089 | 43.2972 | 440.1128 |
| | SoftAdaClip | 0.7363 | 0.7101 | 568.3838 | 43.0088 | 429.5628 |

## C.1 IMPROVEMENTS IN AVERAGE DISPARITY

To further quantify fairness improvements, we compute the **average subgroup disparity** for each dataset and method. For eICU and Adult Income, this metric is defined as the mean of the gender and age disparities, while for MIMIC it includes gender, age, and ethnicity. Each disparity value corresponds to the absolute difference in loss between two subgroups, averaged across five seeds. We then measure the percentage reduction in average disparity achieved by SoftAdaClip relative to DPSGD and Adaptive-DPSGD, using the following formulae:

$$\text{Reduction}_{\text{DPSGD}}(\%) = \frac{\text{AvgDisparity}_{\text{DPSGD}} - \text{AvgDisparity}_{\text{SoftAdaClip}}}{\text{AvgDisparity}_{\text{DPSGD}}} \times 100$$

$$\text{Reduction}_{\text{Adaptive}}(\%) = \frac{\text{AvgDisparity}_{\text{Adaptive}} - \text{AvgDisparity}_{\text{SoftAdaClip}}}{\text{AvgDisparity}_{\text{Adaptive}}} \times 100$$

As summarized in Table 2, SoftAdaClip reduces average subgroup disparities by 25–87% compared to DPSGD and by 8–48% compared to Adaptive-DPSGD across most datasets, with the exception of Income Simple at $C = 0.1$, where disparities slightly increase. These results provide clear evidence that SoftAdaClip consistently improves fairness over baseline methods while preserving formal privacy guarantees.

Table 2: Percentage reduction in average disparity for SoftAdaClip compared to DPSGD and Adaptive-DPSGD. Positive values indicate fairness improvements, while negative values indicate worsening of disparities.

| Dataset | Reduction vs. DPSGD (%) | Reduction vs. Adaptive (%) |
|---------|-------------------------|----------------------------|
| eICU ($C = 0.1$) | 52.7 | 47.9 |
| Income Simple ($C = 0.01$) | 87.3 | 16.5 |
| Income Simple ($C = 0.05$) | 78.5 | 8.4 |
| Income Simple ($C = 0.1$) | -11.7 | -9.5 |
| Income Complex ($C = 0.1$) | 37.5 | 25.9 |
| MIMIC ($C = 0.1$) | 25.2 | 12.5 |

## C.2   Disentangling the Effects of Smoothing and Adaptivity

To isolate the impact of smoothing, we evaluated a variant of SoftAdaClip that uses the tanh-based clipping function with a fixed threshold, and we removed the adaptive component entirely. We refer to this method as *Fixed Soft Clipping*, which applies smooth gradient scaling within the standard DP-SGD framework. This controlled comparison helps determine whether improvements stem from smoothing alone or its combination with adaptivity. We tested this variant across all datasets using five different random seeds and averaged the results to ensure robustness. As shown in Table 3, which reports subgroup loss values (e.g., Male/Female, Age $<$ Median/Age $\geq$ Median, White/Non-White) and their absolute disparities, Fixed Soft Clipping improved fairness over DPSGD in only 6 out of 13 cases and increased the loss gap in 7.

---

**Algorithm 3** FIXED SOFT CLIPPING

**Require:** Iterations $T$, Dataset $D$, sampling rate $q$, clipping bound $C_0$, noise multiplier $\sigma$, learning rates $\eta_t$, small constant $\varepsilon$
**Initialize:** $\theta_0$ randomly
1: **for** $t = 0, \ldots, T-1$ **do**
2:     $B \leftarrow$ Poisson sample of $D$ with rate $q$
3:     **for** each $(x_i, y_i) \in B$ **do**
4:         $g_i \leftarrow \nabla_\theta \ell(f_\theta(x_i), y_i)$
5:         # Smooth clipping via tanh
6:         $\alpha_i \leftarrow \tanh\left(\frac{C_0}{\|g_i\|+\varepsilon}\right)$
7:         $\bar{g}_i \leftarrow \alpha_i \cdot g_i$
8:     **end for**
9:     $\tilde{g}_B \leftarrow \frac{1}{|B|}\left(\sum_{i \in B} \bar{g}_i + \mathcal{N}(0, \sigma^2 C_0^2 I)\right)$
10:     $\theta_{t+1} \leftarrow \theta_t - \eta_t \tilde{g}_B$
11: **end for**

---

In contrast, **SoftAdaClip** consistently achieves smaller subgroup disparities in 11 out of 13 settings, outperforming both Fixed Soft Clipping and Adaptive Clipping. These results indicate that smoothing alone is not reliably effective and highlight that the combination of smooth and adaptive clipping, as used in SoftAdaClip, is key to improving fairness.

Table 3: Full experimental results across datasets and methods. Losses and subgroup disparities are averaged across all seeds. Group Difference refers to the absolute difference in loss between the two subgroups within each demographic attribute (e.g., male vs. female for gender, below vs. above median for age, White vs. non-White for ethnicity).

| Dataset | Method | Male Loss | Female Loss | Gender Diff | Age < Median | Age ≥ Median | Age Diff | White Loss | Non-White Loss | Ethnicity Diff |
|---|---|---|---|---|---|---|---|---|---|---|
| eICU (C = 0.1) | Non-Private | 9.5598 | 9.3059 | 0.2539 | 8.6992 | 10.1524 | 1.4532 | – | – | – |
| | DPSGD | 48.5335 | 48.8912 | 2.2972 | 43.1970 | 53.7035 | 10.5065 | – | – | – |
| | Adaptive | 49.2247 | 49.5550 | 1.3555 | 43.9941 | 54.2443 | 10.2502 | – | – | – |
| | SoftAdaClip | 25.8796 | 25.7997 | 0.7224 | 23.0494 | 28.3781 | 5.3287 | – | – | – |
| | Fixed Soft Clipping | 48.4679 | 48.451 | 2.4615 | 43.2143 | 53.3018 | 10.0875 | | | |
| Income Simple (C = 0.1) | Non-Private | 15.1112 | 8.4555 | 6.6557 | 11.8803 | 16.0424 | 4.1621 | – | – | – |
| | DPSGD | 151.5774 | 95.0878 | 56.4896 | 121.2306 | 164.9877 | 43.7572 | – | – | – |
| | Adaptive | 139.9777 | 81.3574 | 58.6203 | 112.3326 | 155.9813 | 43.6487 | – | – | – |
| | SoftAdaClip | 202.1400 | 112.3267 | 89.8133 | 177.8177 | 197.2410 | 22.2014 | – | – | – |
| | Fixed Soft Clipping | 412.4973 | 325.84348 | 103.11782 | 379.45258 | 464.71968 | 85.2671 | | | |
| Income Simple (C = 0.01) | DPSGD | 560.8699 | 184.2400 | 376.6299 | 357.6398 | 630.7931 | 273.1533 | – | – | – |
| | Adaptive | 146.6832 | 92.9323 | 53.7509 | 119.5134 | 164.2264 | 44.7130 | – | – | – |
| | SoftAdaClip | 143.7968 | 98.9942 | 44.8026 | 121.4856 | 158.8987 | 37.4131 | – | – | – |
| | Fixed Soft Clipping | 554.227 | 182.5821 | 371.6449 | 353.8296 | 623.2183 | 269.3887 | | | |
| Income Simple (C =0.05) | DPSGD | 388.9525 | 154.7919 | 234.1606 | 272.2588 | 433.8648 | 161.6060 | – | – | – |
| | Adaptive | 137.3956 | 91.1264 | 46.2692 | 113.2906 | 159.9958 | 46.7052 | – | – | – |
| | SoftAdaClip | 136.3423 | 93.3233 | 43.0190 | 114.2903 | 156.3899 | 42.0996 | – | – | – |
| | Fixed Soft Clipping | 352.1058 | 145.5571 | 206.5487 | 249.7627 | 393.5827 | 143.82 | | | |
| Income Complex (C = 0.1) | Non-Private | 15.3451 | 8.8033 | 6.5418 | 12.2018 | 16.2263 | 4.0245 | – | – | – |
| | DPSGD | 176.1793 | 97.9874 | 78.1918 | 134.4287 | 179.7389 | 45.3102 | – | – | – |
| | Adaptive | 147.5599 | 83.8232 | 63.7367 | 111.3624 | 151.8005 | 40.4382 | – | – | – |
| | SoftAdaClip | 111.4720 | 63.9662 | 47.5059 | 84.0589 | 113.7171 | 29.6582 | – | – | – |
| | Fixed Soft Clipping | 176.3323 | 98.0936 | 78.2387 | 131.8692 | 182.0531 | 50.1839 | | | |
| MIMIC (C = 0.1) | Non-Private | 12.9760 | 12.9520 | 0.0240 | 13.0120 | 12.3640 | 0.6480 | 12.6360 | 12.7920 | 0.1560 |
| | DPSGD | 822.0909 | 812.0865 | 48.6868 | 803.2820 | 829.0804 | 31.9105 | 821.5653 | 829.0350 | 33.5979 |
| | Adaptive | 597.0828 | 562.5361 | 34.5467 | 590.0785 | 565.6026 | 28.4748 | 571.1686 | 550.5871 | 34.6889 |
| | SoftAdaClip | 574.3803 | 565.3295 | 29.1806 | 579.0692 | 545.1397 | 33.9295 | 566.2895 | 549.9928 | 22.3648 |
| | Fixed Soft Clipping | 813.5449 | 903.4000 | 95.6308 | 816.0311 | 794.9747 | 42.8099 | 809.8198 | 799.3655 | 15.9609 |

## C.3   Gradient Norm Clipping Results

Table 4 reports the average gradient norms before and after clipping for each subgroup. Figure 6 presents the same information in a visual form, providing a clearer illustration of the differences across groups. As discussed in the main text, SoftAdaClip applies less clipping than other methods due to its smooth transformation. Even Fixed Soft Clipping results in lower clipping than standard DP-SGD. An exception is the Income Simple dataset with $C = 0.1$, where the gradients are already small, leading to minimal clipping under DP-SGD. In this setting, SoftAdaClip initially offered limited benefit, as there was little suppression to improve upon. However, when we reduced the

clipping threshold, the gradients became more likely to be clipped, and SoftAdaClip demonstrated its advantage by smoothing the clipping operation and improving fairness.

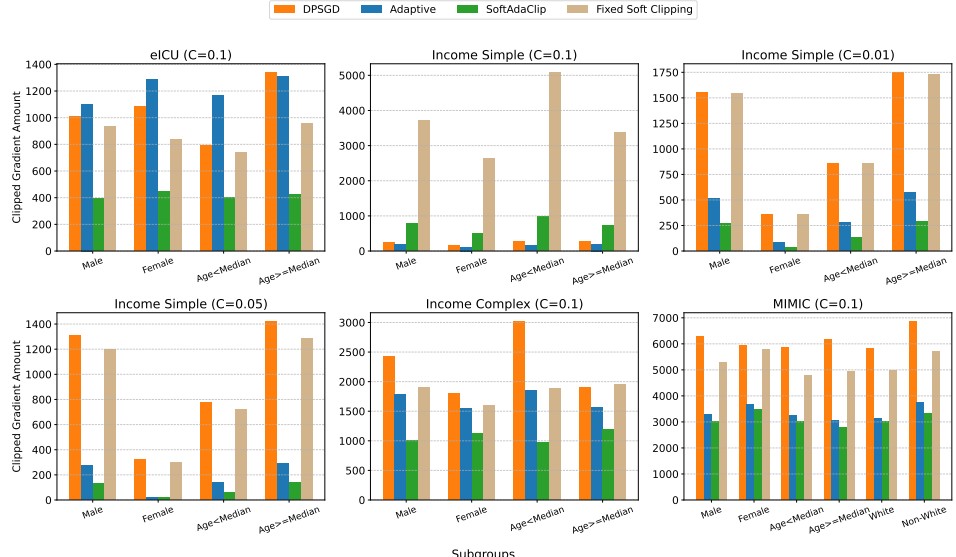

Figure 6: Histogram of clipped gradient amounts across demographic groups for each dataset and method. Bars represent the average gradient being clipped for each subgroup under DPSGD, Andrew et al. (2021) Adaptive Clipping, SoftAdaClip, and Fixed Soft Clipping. Results are shown separately for eICU, Income (Simple and Complex, with varying $C$), and MIMIC. Across all datasets, SoftAdaClip consistently clips substantially less than the other approaches when the clipping threshold is set appropriately, indicating that it preserves more subgroup signal while still respecting privacy constraints.

Table 4: Gradient statistics across all datasets and methods. Each cell shows: Before → After (Before After).

| Dataset | Method | Male | Female | Age < Median | Age >= Median | White | Non-White |
|---|---|---|---|---|---|---|---|
| eICU ($C = 0.1$) | DPSGD | 1005.86→0.48 (1005.38) | 1082.48→0.49 (1081.99) | 794.95→0.46 (794.50) | 1337.97→0.52 (1337.45) | – | – |
| | Adaptive | 1102.94→0.50 (1102.45) | 1289.88→0.50 (1289.38) | 1169.67→0.46 (1169.21) | 1309.35→0.53 (1308.82) | – | – |
| | SoftAdaClip | 403.13→5.85 (397.27) | 456.32→9.37 (446.95) | 405.55→5.69 (399.85) | 432.38→9.73 (422.65) | – | – |
| | Fixed Soft Clipping | 930.94→0.48 (930.46) | 836.88→0.49 (836.39) | 742.72→0.45 (742.27) | 954.59→0.51 (954.08) | – | – |
| Income Simple ($C = 0.1$) | DPSGD | 254.58→0.56 (254.02) | 152.37→0.43 (151.94) | 260.21→0.54 (259.67) | 261.84→0.62 (261.21) | – | – |
| | Adaptive | 182.82→0.60 (182.21) | 114.82→0.46 (114.35) | 153.92→0.56 (153.36) | 200.79→0.66 (200.14) | – | – |
| | SoftAdaClip | 773.40→0.58 (772.82) | 500.16→0.42 (499.74) | 990.73→0.51 (990.22) | 719.96→0.66 (719.30) | – | – |
| | Fixed Soft Clipping | 3709.43→0.54 (3708.88) | 2644.33→0.39 (2643.94) | 5075.98→0.50 (5075.48) | 3368.97→0.65 (3368.32) | – | – |
| Income Simple ($C = 0.01$) | DPSGD | 1555.11→0.58 (1554.53) | 362.81→1.74 (361.07) | 862.32→1.13 (861.20) | 1746.93→0.55 (1746.38) | – | – |
| | Adaptive | 516.74→1.36 (515.39) | 89.68→8.40 (81.28) | 280.38→0.86 (279.52) | 568.9→2.98 (568.92) | – | – |
| | SoftAdaClip | 268.24→1.43 (266.81) | 39.62→8.22 (31.41) | 134.50→1.89 (132.61) | 293.66→3.15 (290.51) | – | – |
| | Fixed Soft Clipping | 1542.81→0.56 (1542.25) | 360.00→1.67 (358.32) | 856.21→1.08 (855.13) | 1732.34→0.53 (1731.80) | – | – |
| Income Simple ($C = 0.05$) | DPSGD | 1311.16→2.25 (1308.91) | 327.27→4.00 (323.27) | 776.91→2.50 (774.41) | 1421.75→2.71 (1419.04) | – | – |
| | Adaptive | 278.38→7.21 (271.17) | 48.60→30.58 (18.01) | 150.86→9.50 (141.36) | 301.32→14.23 (287.09) | – | – |
| | SoftAdaClip | 134.88→7.11 (127.77) | 53.1→29.36 (23.73) | 72.0→8.92 (63.1) | 157.01→14.66 (142.34) | – | – |
| | Fixed Soft Clipping | 1198.26→2.29 (1195.97) | 303.94→4.03 (299.91) | 718.75→2.49 (716.26) | 1289.85→2.74 (1287.11) | – | – |
| Income Complex ($C = 0.1$) | DPSGD | 2420.41→0.96 (2419.45) | 1800.54→0.78 (1799.76) | 3014.40→0.70 (3013.70) | 1905.54→1.08 (1904.46) | – | – |
| | Adaptive | 1780.67→1.21 (1779.46) | 1541.20→1.56 (1539.64) | 1847.32→1.11 (1846.21) | 1572.68→1.34 (1571.34) | – | – |
| | SoftAdaClip | 1003.58→4.58 (998.99) | 1126.95→6.86 (1120.09) | 976.81→3.44 (973.38) | 1201.06→6.50 (1194.56) | – | – |
| | Fixed Soft Clipping | 1907.96→0.67 (1907.29) | 1589.68→0.74 (1588.94) | 1886.86→0.61 (1886.25) | 1954.12→0.70 (1953.42) | – | – |
| MIMIC ($C = 0.1$) | DPSGD | 6276.55→1.82 (6274.73) | 5943.33→2.35 (5940.99) | 5847.74→1.94 (5845.79) | 6151.16→1.92 (6149.24) | 5810.35→1.97 (5808.38) | 6854.15→1.92 (6852.23) |
| | Adaptive | 3286.83→2.55 (3284.28) | 3669.94→2.50 (3667.44) | 3233.49→3.39 (3230.11) | 3073.35→2.63 (3070.71) | 3150.46→2.89 (3147.58) | 3748.57→3.42 (3745.15) |
| | SoftAdaClip | 3019.59→3.02 (3016.57) | 3486.80→2.47 (3484.34) | 3018.68→4.60 (3014.08) | 2802.01→2.91 (2799.09) | 3012.22→3.36 (3008.86) | 3346.56→4.93 (3341.63) |
| | Fixed Soft Clipping | 5302.20→1.77 (5300.42) | 5776.64→2.23 (5774.41) | 4800.18→1.85 (4798.32) | 4941.69→1.82 (4939.87) | 4995.37→1.86 (4993.51) | 5726.17→1.87 (5724.30) |

# D NOTE ON LANGUAGE EDITING

For the preparation of this paper, we used ChatGPT to improve the clarity, grammar, and style of the writing. This tool was specifically employed for the purposes of refining and editing the text; it was not used to generate ideas, content, or contribute to the research itself.

