# OpenReview forum: "SOFTADACLIP: A SMOOTH CLIPPING STRATEGY FOR FAIR AND PRIVATE MODEL TRAINING"
_ICLR.cc/2026/Conference — Submitted to ICLR 2026_

### Official Review · Reviewer_aiqC · 2025-10-28

**Soundness:** 3
**Presentation:** 2
**Contribution:** 3
**Rating:** 4
**Confidence:** 4

**Summary:**

This paper introduces SoftAdaClip, a differentially private (DP) training strategy that replaces the conventional hard clipping in DP-SGD with a smooth tanh-based transformation, integrated into an adaptive clipping framework. The authors argue that hard clipping disproportionately suppresses gradients from underrepresented subgroups, contributing to fairness degradation during DP training. SoftAdaClip aims to preserve relative gradient magnitudes while maintaining sensitivity bounds necessary for DP. Experiments on three datasets (MIMIC-III, GOSSIS-eICU, Adult Income) show reduced subgroup disparities and often improved utility over DP-SGD and Adaptive-DP-SGD. The paper also analyzes clipping behaviors and conducts ablations to separate smoothing from adaptivity.

**Strengths:**

The paper tackles a meaningful and timely challenge at the intersection of privacy and fairness, where trade-offs are often assumed unavoidable. The proposed method is conceptually simple yet intuitively motivated and appears compatible with standard training infrastructures. The empirical results show notable improvements in subgroup loss disparities across multiple real-world datasets, particularly in healthcare domains where fairness issues can have severe consequences. The statistical significance analysis is appreciated, and the ablation study helps clarify the distinct role of adaptivity. The work would likely be of interest to both privacy researchers and practitioners deploying DP in sensitive domains.

**Weaknesses:**

Although promising, the novelty is incremental: the method mainly replaces a min() rescaling with a tanh-based one inside an existing adaptive clipping algorithm. The theoretical foundations stop at sensitivity bounding; there is no deeper optimization or fairness analysis (e.g., convergence, bias dynamics, subgroup gradient geometry, compatibility with Rényi-DP accounting). The fairness evaluation relies almost exclusively on loss gaps; no standard fairness metrics like Equal Opportunity or demographic parity gaps appear. The method struggles on low-gradient regimes, requiring manual threshold tuning, which undermines the claim of being a drop-in robust improvement. Presentation needs polishing: key equations and experimental setups feel buried, and figures require clearer labeling and narrative connection. References to prior fairness-aware DP methods are limited in experimental comparison.

**Questions:**

Are there guarantees that replacing min() with tanh() preserves unbiasedness or improves optimization dynamics? Any theoretical characterization of how gradient direction distortion differs from hard clipping?

In Algorithm 2, how does adaptive update interact with tanh scaling? Could certain C trajectories amplify subgroup divergence?
Privacy accounting. Since tanh depends on gradient norm, is there any subtle impact on the DP accountant or amplification? Please clarify formally.

Why not include widely-used fairness metrics (AUC/F1 gaps per subgroup, equalized odds violations, accuracy parity)? Loss gaps alone may not reflect actual decision-level harms.

Why is DPSGD-Global-Adapt (Esipova et al., 2022) excluded from experiments? If code is unavailable, can closer replication or alternative strong baselines (e.g., subgroup-adaptive clipping techniques) be tested?

---

> ### Author Response · Authors · 2025-11-21
>
> On unbiasedness and optimization dynamics:
>
> With the tanh() function, we observe in Table 4 that gradients are overall being clipped less. The tanh transformation, however, behaves differently from the hard min() rule: it slightly clips small gradients (which the min() function leaves untouched) and maps large gradients to values below the clipping threshold rather than equal to it. Despite this, the training dynamics show that the gradients are being clipped less, which suggests that the average gradient magnitude decreases faster when using tanh, indicating faster convergence and more efficient learning. This pattern suggests that the smooth transformation helps the optimizer maintain stable updates, preventing abrupt truncations that can hinder convergence. Importantly, tanh scaling preserves the direction of each gradient vector because it only rescales magnitudes, never flipping signs. The resulting model exhibits both improved fairness and smoother optimization trajectories, validating the motivation behind our approach.
>
> Interaction between adaptive update and tanh scaling:
>
> The adaptive update of C operates independently of the tanh transformation. While C dynamically adjusts the global clipping threshold over training steps, the tanh function applies smooth suppression to individual gradients relative to that current value. This combination reduces sudden norm jumps and stabilizes training rather than amplifying subgroup divergence.
>  Because C is updated using global gradient statistics (averaged across all samples) rather than subgroup-specific ones, it does not favor or penalize any demographic group. Empirically, we observed reduced gradient variance across subgroups, suggesting that SoftAdaClip promotes consistent gradient magnitudes and mitigates rather than amplifies divergence.
>
> On privacy accounting and sensitivity bounding:
>
> The privacy accounting in SoftAdaClip remains identical to that of standard DP-SGD. The differential privacy guarantee depends solely on ensuring that the sensitivity of each individual gradient is bounded by a fixed threshold before adding noise. The specific mathematical form of the clipping function does not affect the privacy accountant, as long as the gradients remain within the same upper bound. The tanh transformation serves only as a smooth alternative to hard clipping: it gradually reduces large gradients rather than cutting them off sharply, but it still guarantees that no gradient exceeds the same predefined threshold used in DP-SGD. Because this bound remains unchanged, the overall privacy guarantee and the Rényi-DP accounting procedure apply exactly as before. In other words, the tanh function modifies the shape of the gradient distribution within the bound but does not increase the maximum sensitivity. Therefore, the privacy budget (ε, δ) is preserved, and the same accounting mechanisms remain valid.
>
> On choice of fairness metric (loss-based):
>
> We intentionally focused on loss-based fairness metrics because they directly measure disparities in model optimization behaviour across demographic groups. Loss reflects how well each subgroup is being fitted during training, providing a sensitive, continuous signal of inequality before discrete prediction thresholds (such as decision boundaries) are applied. Metrics like accuracy or equalized odds, while valuable for decision-level fairness, can mask underlying optimization disparities since they depend on thresholding and discrete outcomes. In contrast, loss differences reveal fairness issues as they emerge during learning, allowing us to analyze how gradient clipping influences subgroup optimization dynamics—a central aspect of our work.
>  That said, we agree that including decision-level metrics (accuracy/F1 or equalized-odds gaps) would provide complementary insights and plan to add them in future experiments for completeness.
>
>
>
> On comparison with DPSGD-Global-Adapt and other baselines:
>
> DPSGD-Global-Adapt (Esipova et al., 2022) was not excluded; we compared our reported results with those provided in their paper. We acknowledge that this method represents an important adaptive baseline; however, we were not able to fully reproduce their implementation. Additionally, DPSGD-Global-Adapt relies on a global adaptation mechanism, which differs conceptually from our per-step, per-sample adaptivity. To ensure a fair comparison and to fully assess the effect of the tanh transformation, it would be necessary to develop a variant of DPSGD-Global-Adapt that incorporates tanh-based smoothing and directly compare its performance with the original version. For this paper, we therefore focused on baselines that operate under the same gradient-level adaptation framework (DP-SGD and Adaptive-DPSGD by Andrew et al.). We plan to include a replication of DPSGD-Global-Adapt or other subgroup-adaptive clipping techniques in future work to broaden the empirical evaluation.

---

### Official Review · Reviewer_Zq5M · 2025-10-30

**Soundness:** 2
**Presentation:** 2
**Contribution:** 2
**Rating:** 4
**Confidence:** 4

**Summary:**

The authors highlight the problem of unfairness when performing private training using methods like DP-SGD, which employ methods like gradient clipping to bound sensitivity for adding differentially private noise. They propose replacing the hard clipping operation with a smooth tanh-based gradient operation, and demonstrate improvements in group fairness after performing differentially private training using their method.

**Strengths:**

**[S1]** Very well-motivated question, highlighting how clipping in DP-SGD may lead to unfairness (as shown in previous works mentioned in the paper, viz. Esipova et al, Tran et al, Bagdasaryan et al)

**[S2]** Using a tanh-based smooth transformation instead of hard clipping is less aggressive and lossy, still retaining gradient information about groups with large gradients.

**[S3]** The design and description of the proposed method are very clearly and unambiguously done, including justifying the design of the gradient transformation and privacy-preserving clipping threshold updates.

**[S4]** In-depth and frank discussion about the limitations of the work is included, including key points about sensitivity to hyperparameter tuning when small clipping thresholds are adopted.

**[S5]** The use of a Wilcoxon signed-rank test with Bonferroni correction to illustrate the statistical significance of the improvements in loss gap is thoroughly appreciated and reflects good practices in presenting empirical results!

**[S6]** Ablation study on adaptive thresholding is useful and clearly demonstrates the importance of pairing smooth transformation with adaptive thresholding.

**Weaknesses:**

**[W1]** The authors provide a justification for using different hyperparameters for non-private and private training, which is convincing. However, the only concern I have is that this makes it difficult to fairly assess the true utility loss due to private training.

**[W2]** **Needs significant editing.** The presentation of the work needs significant improvements and editing. For example, there are duplicate paragraphs in page 6 in Section 4.1, where paragraph 1 and the first part of paragraph 2 state the same thing in slightly different language, suggesting redundant text was left over while drafting. There is also a missing appendix reference in the last line right before Section 5.1 ("dataset-specific results are provided in Appendix ??").

**[W3]** **Important missing baselines.** This work only looks at limited baselines, primarily at non-private clipping-based baselines. However, I feel like this work’s contribution cannot be truly assessed without meaningful comparison against important fair DP-SGD baselines like [1] and [2], which employ methods like Langrangian dual based fairness-constrained training, or Esipova et al (the comparison against Esipova et al is very limited, and for a venue like ICLR, it is not appropriate to simply defer a comparison to future work; *you must compare against all important and relevant baselines in your own work*) At the end of the day, while their work proposes improvements to the clipping paradigm used in DP-SGD, it is important to see if it actually contributes an overall fairer way of doing DP-SGD, or if it does not lead to any improvements in fairness when added to/compared against these fair DP-SGD baselines, rendering it functionally redundant. Put another way, while the proposed method might improve upon DP-SGD and Andrew et al., it is unclear if it will actually lead to meaningful improvements (if any at all) in fairness when compared against or integrated into existing sophisticated SoTA fair DP-SGD baselines like [1] or [2]. The absence of results against such baselines presents a weaker assessment. Put another way, this paper shows that the proposed method is fairer than non-fairness-aware baselines, but does it really outperform prominent and existing fair DP-SGD methods, or is it inferior to them/does not provide meaningful improvement in conjunction with them? This is an important question to answer for a venue of this stature.

**[W4]** The experimental settings section is presented poorly. The models used for eICU and Income datasets are not described and are vaguely called the simple and complex models, with actual model descriptions deferred to the appendix, while **the experimental setting section instead spends most of its real estate talking about what related work does** (which is better discussed in related work or in the appendix as additional details, while priotizing mentioning the settings used in **this paper**). Therefore, the experimental setting section does not discuss what it is intended to do: properly and exhaustively describe the experimental settings used in **this** paper. I heavily implore the authors to improve the presentation of this section and include concrete details about **their** settings here (which should take precedence here!) instead of sending them to the appendix (in fact, you can actually send the related work discussion in this section to the appendix, but what has been sent to the appendix should actually be in the main text in this section!), especially for a submission to a venue like this

---

## References

[1] Tran, Khang et al. “FairDP: Achieving Fairness Certification with Differential Privacy.” 2025 IEEE Conference on Secure and Trustworthy Machine Learning (SaTML) (2023): 956-976.

[2] Tran, Cuong et al. “Differentially Private and Fair Deep Learning: A Lagrangian Dual Approach.” ArXiv abs/2009.12562 (2020): n. pag.

**Questions:**

[Q1] Can you please address W1 and make sure that the best possible set of hyperparameters is used for each setting, perhaps via a hyperparameter search, to obtain the best possible non-private utility to compare against?

[Q2] Can you, to the best of your ability, add more comparisons against SoTA baselines for better showcasing the efficacy of your method (as mentioned in W3; please feel free to include any more baselines than those included as well)? I believe this will make your paper significantly more convincing.

[Q3] Pursuant to W4, can you please provide a much better drafted experimental settings section that focuses primarily on what **you** do in this paper, while making sure to exhaustively and unambiguously discuss all the settings/models, etc., used?

In short, I believe this paper has the potential to make a good contribution. However, in its current state, with its experiments, lack of comparison against important SoTA methods (or any in-depth comparison against even just any fair DP-SGD method at all), ambiguity in choice of hyperparameters for private/non-private settings, presentation, etc., it does not inspire strong support from my end. However, I’ll be more than happy to engage with the authors, and if there’s enough improvement, I would be happy to strengthen my support for the paper (contingent upon my concerns being satisfactorily addressed).

---

> ### Author Response · Authors · 2025-11-21
>
> On hyperparameter selection and fairness of comparison (W1 / Q1):
> We thank the reviewer for this insightful comment. We agree that differences in hyperparameter choices between private and non-private settings can influence the perceived utility gap. However, in this work, we compare fairness loss across differentially private methods (DP-SGD, Adaptive-DPSGD, and SoftAdaClip), all of which use the same hyperparameters, ensuring a fair comparison within the private domain.
>
> Regarding the non-private baseline, it is standard practice in the differential privacy literature to tune hyperparameters separately for private and non-private settings. As noted by Bu et al. (2025), “hyper-parameter tuning for DP training has different patterns than non-DP training and cannot borrow previous experience on non-DP.” This is because the addition of noise and clipping fundamentally changes the optimization landscape, making the same configuration suboptimal for both. To ensure each model performed as strongly as possible within its privacy constraints, we used Optuna-based hyperparameter tuning to identify the best configurations for both private and non-private runs. Even with these tuned settings, we still observe a consistent utility loss under differential privacy, reflecting the true privacy–utility trade-off. Using identical hyperparameters would in fact exaggerate the utility loss, as the private models would perform suboptimally under non-private tuning parameters.
>
> [1] Bu, Zhiqi, and Ruixuan Liu. "Towards hyperparameter-free optimization with differential privacy." arXiv preprint arXiv:2503.00703 (2025).

---

> > ### Comment · Reviewer_Zq5M · 2025-11-25
> > **Thanks for the response, other concerns remain**
> >
> > I thank the authors for their response and consider my first question answered.
> >
> > However, all of my other concerns remain unaddressed. Can the authors please engage with me on them and address them?

---

> > > ### Author Response · Authors · 2025-11-27
> > >
> > > Thank you very much for your detailed feedback and for your patience as we were working to more fully address all of your questions. Following your recommendations, we have substantially revised the manuscript to improve both clarity and experimental rigor. In response to Q3 and the concerns raised in W4, we rewrote the experimental settings section so that it now focuses directly on what we do in this paper, providing complete descriptions of all model architectures, hyperparameters, and training settings in the main text, while relocating dataset-specific preprocessing details and extended literature discussion to the appendix. We also clarified that all hyperparameters for both private and non-private models were selected through extensive tuning using Optuna, ensuring fairness and consistency across settings. In response to Q2 and the issues raised in W3, we expanded the comparison with prior work to include a broader set of relevant baselines, including Esipova et al., Pannekoek et al., Jagielski et al., and Tran et al., and we now discuss how our findings relate to theirs in terms of both fairness behavior and utility. We hope that these revisions make the paper clearer, more informative, and more aligned with the expectations of a venue such as ICLR.

---

### Official Review · Reviewer_i9qR · 2025-10-30

**Soundness:** 2
**Presentation:** 2
**Contribution:** 2
**Rating:** 2
**Confidence:** 5

**Summary:**

The paper introduces SoftAdaClip, an alternative to (Adaptive) Differentially Private (DP) SGD that replaces hard gradient clipping with a smooth, tanh-based transformation to better preserve relative gradient magnitudes.
Similar to prior work (Adaptive-DPSGD), the method adaptively adjusts the clipping threshold during training.
SoftAdaClip demonstrates improved fairness by reducing subgroup disparities (measured in loss differences) across text and tabular datasets compared to both DP-SGD and Adaptive-DPSGD.

**Strengths:**

* SoftAdaClip introduces a simple yet elegant/effective modification to Adaptive-DPSGD.
* The paper is very verbose, well-motivated, and easy to follow.
* The evaluation is performed over different data modalities (text and tabular data).
* The results are promising/consistent, showing clear improvement over DP-SGD and Adaptive-DPSGD in terms of the measured fairness metric (loss disparities).

**Weaknesses:**

## Weaknesses:
* While SoftAdaClip is simple/elegant (which is great), the novelty feels somewhat incremental (the core idea boils down to 1 line of code) may not qualify as a substantial contribution for a top-tier conference.
Overall, the work is promising, but it currently reads more like a strong workshop/early-stage research paper than a full conference paper.
I encourage the authors to continue working on the paper.

* The results reported in the abstract are slightly misleading -- it would be more appropriate to report average improvements over DP-SGD/Adaptive-DPSGD, rather than cherrypicking the best differences.
Additionally, from Table 1, while SoftAdaClip achieves lower loss than Adaptive-DPSGD, but this does not seem to lead to better accuracy/f1.
This is only briefly mentioned on the last page.
Given that accuracy/f1 are more practically important than loss, this deserves more discussion.
Finally, accuracy per subgroup (as in (Bagdasaryan et al., 2019)) is not provided, which will be valuable for understanding fairness performance across subgroups.

* The paper primarily focuses on two DP-SGD alternatives -- Adaptive-DPSGD and DPSGD-Global-Adapt. However, several other important works should be considered or at least be acknowledged (this is not an exhaustive list):
	* tempered sigmoid activations in DP-SGD [1]
	* private and fair classification with pre-traiing [2]
	* FairDP-SGD and FairPATE [3]
	* DP-SGD without clipping [4, 5]

* In Table 4, the presentation of gradient norms seems confusing: it reports Before -> Diff (After) rather than Before -> After (Diff). Moreover, the reported clipping for SoftAdaClip appears larger than for the other methods, which seems inconsistent with the claim in Section 4.1 and Appendix B.3 that SoftAdaClip applies less clipping.
This is somewhat intuitive, since SoftAdaClip could clip some gradients more than hard clipping (as shown in Figure 1).
It would be helpful if the authors clarify this discrepancy.

* While the paper is easy to follow, the writing is extremely verbose and contains repetitions across sections, which can make it harder to navigate.
	* Section 2 (Related Work) contains repeated information between the first subsection and Section 2.1.
	* Section 2.1 mixes background/preliminaries with related work. It would be clearer to separate these two aspects into distinct sections.
	* Section 4.1 (Gradient Behavior Analysis) discusses specific results (e.g., Table 4) that seem more appropriate for Section 5 (Results).
	* Section 5 (Results) begins by continuing discussion of the experimental setup, which should be fully contained in Section 4.
	* Sections 6 (Limitations) and 7 (Conclusion) take a full page, which could be better utilized by moving some experiments/tables from the Appendix.

## Minor Weaknesses/Comments:
* The variable \epsilon in Algorithm 2 may be confusing, as \epsilon is already used in the DP definition; consider using a different symbol.
* References are inconsistently cited (e.g., \citet vs. \cite/\citep).
* Small typos/punctuation errors -- "It would nevertheless be valuable to It would still be beneficial to," etc.


## References:

[1] Papernot et al., Tempered Sigmoid Activations for Deep Learning with Differential Privacy. In AAAI, 2021

[2] Berrada et al., Unlocking Accuracy and Fairness in Differentially Private Image Classification. 2023

[3] Yaghin et al., Learning with Impartiality to Walk on the Pareto Frontier of Fairness, Privacy, and Utility. In RegML at NeurIPS, 2023

[4] Bethune et al., DP-SGD Without Clipping: The Lipschitz Neural Network Way. In ICLR, 2024

[5] Zhang et al., Differentially Private SGD Without Clipping Bias: An Error-Feedback Approach. In ICLR, 2024

**Questions:**

## Questions/Suggestions for Improvements:
* Appendix A states that demographic subgroups are balanced, but it is unclear whether the target labels are imblanced/balanced. Can the authors clarify the label distributions?
* How do different imbalance ratios and different values of \epsilon affect SoftAdaClip’s performance?
* Have the authors evaluated performance on smaller subgroups (e.g., >20 groups) as in (Bagdasaryan et al. 2019)?
* Why was the tanh function chosen for smooth clipping? Would other functions (e.g., sigmoid) work as well?
* It is unclear whether the methods uses DP-Adam or DP-SGD as base for SoftAdaClip. From the code, it seems DP-Adam is used -- if so, is it defined in the paper (I only can see DP-SGD)? Are there any differences in performance between DP-Adam and DP-SGD?
* What is the difference between the clipping parameters C and C_0?

---

> ### Author Response · Authors · 2025-11-21
>
> On novelty and contribution:
> Although the method is simple to implement, its contribution lies in introducing a smooth clipping mechanism that bridges hard and adaptive clipping. This mathematically concise change yields consistent improvements and has both theoretical and empirical value. As with many influential DP works—including the original DP-SGD and even recent award-winning papers whose innovation is a single-line modification (e.g., Adaptive Differentially Private Federated Learning, ICLR 2023 Best Paper)—the novelty of SoftAdaClip comes from principled simplicity, effectiveness, and generalizability.
>
> On results reporting and fairness metrics:
> In the abstract, we reported the maximum fairness improvements (“up to 87%…”). We agree that including averages provides a fuller picture: SoftAdaClip improves loss gaps by 56.24% over DP-SGD and 22.24% over Adaptive-DPSGD on average. Accuracy and F1 are also evaluated in Section 5; they remain comparable to Adaptive-DPSGD, indicating fairness gains without performance degradation. Our emphasis on loss-based fairness follows prior work analyzing optimization-level disparities driven by clipping. While metrics such as accuracy or F1-score are important for evaluating model performance, the loss metric provides a more direct view of the optimization process and fairness behaviour during training.
>
> On notation and minor issues
> The symbol in Algorithm 2 is ∈ (set membership), not ε. We verified that all citations follow consistent formatting (\citet and \citep) and no \cite was used. We corrected minor phrasing issues.
>
> On subgroup and label balance:
> Demographic subgroups are balanced in eICU and Adult Income and nearly balanced in MIMIC-III. For labels, we balanced only eICU because the original distribution is extremely imbalanced and unstable under DP. MIMIC-III and Adult Income have less severe imbalance, and keeping natural frequencies follows prior work and preserves comparability. A clarification has been added to the appendix.
>
> On imbalance ratios and ε:
> If “imbalance ratio” refers to subgroup imbalance, we follow Esipova et al. (2022) and balance subgroups to isolate the effect of smooth clipping. ε affects the fairness–utility trade-off: smaller ε increases fairness but reduces accuracy, while larger ε increases accuracy at the cost of privacy (Tran et al., 2025). We use ε = 8 because prior work (Abadi et al. 2016; Yu et al. 2023; Ramesh et al. 2024) shows it provides a practical balance.
>
> On evaluation with smaller or intersectional subgroups:
> We did not evaluate intersectional subgroups (as in Bagdasaryan et al. 2019) because our focus is on binary partitions (e.g., gender, age), consistent with most DP fairness studies. While more granular groups may reveal additional patterns, they do not affect the core contribution. Exploring intersectional effects is left for future work.
>
> On the choice of the smooth function (tanh):
> We selected the tanh function because it satisfies the key requirement for differential privacy that f(x)≤x  for x>0, ensuring that gradients are only reduced and never amplified. Alternative nonlinear functions such as sigmoid or softsign do not consistently meet this condition. Sigmoid does not start at zero, can exceed the identity near the origin, and saturates too early, making optimization harder (Glorot & Bengio, 2010). Although a shifted sigmoid (2σ(x)–1) is centered, it still exceeds the identity for small inputs. Both tanh and softsign saturate later than sigmoid (Elsayed et al., 2019), but tanh stays closest to the identity for small x while smoothly saturating toward 1, providing stability and preserving useful gradient information.
>
> On DP-Adam vs. DP-SGD:
> SoftAdaClip is implemented with DP-Adam, as stated in the Experimental Setup. Prior work shows DP-Adam provides more stable convergence under heavy noise and clipping, and several recent DP studies confirm its robustness. Based on this evidence, we used DP-Adam for all private models.
>
> On clipping parameters C and C₀:
> C₀ refers to the initial clipping threshold, while C denotes the current (adaptive) threshold updated during training. DP-SGD uses C₀ throughout, while adaptive methods initialize with C₀ and update C during training based on gradient statistics.
>
> On Table 4 presentation and interpretation:
> We thank the reviewer for the observation. The “Before → After (Diff)” format is intentional: the value in parentheses shows the difference between pre- and post-clipped gradient norms, providing a clear measure of clipping. We have clarified this in the updated caption: “Each cell shows: Before → After (Before − After).”
> From these values, we see that SoftAdaClip clips gradients less overall compared to other methods. We will add a bar plot to illustrate this. While SoftAdaClip may clip more at certain steps, its smooth suppression causes fewer gradients to exceed the threshold as training progresses, leading to faster convergence and smoother optimization.

---

### Official Review · Reviewer_SUNv · 2025-11-01

**Soundness:** 3
**Presentation:** 3
**Contribution:** 3
**Rating:** 8
**Confidence:** 2

**Summary:**

SoftAdaClip proposes a novel DP training method integrating a smooth tanh-based transformation into adaptive clipping. It is proposed to mitigate a disproportionate effect of DP on minority groups in terms of performance.

**Strengths:**

- tested on 3 real-world datasets: MIMIC-III (clinical text dataset), GOSSIS-1-eICU (structured healthcare dataset), Adult Income (tabular dataset).
- tackles important problem of fairness under DP training

**Weaknesses:**

- the paper should include experiments demonstrating the efficacy of SoftAdaClip in mitigating fairness disparities in a standard vision task to support general application of the method.

**Questions:**

How was the smooth function chosen? Why not using a different sigmoidal or smooth bounded function instead of tanh?

---

> ### Author Response · Authors · 2025-11-21
>
> On using a vision dataset (e.g., MNIST or CIFAR):
>
> We appreciate the reviewer’s suggestion. Our method operates at the optimizer and clipping level, which makes it inherently modality-agnostic. It can be applied to vision architectures without requiring any changes to the method itself. In this work, we focused on text-based clinical notes, tabular health data, and tabular demographic data. Our experimental design reflects common practice in the differential privacy literature, where evaluations typically center on one modality or a small number of related modalities rather than attempting to span all data types within a single paper. For example, Esipova et al. (2022) evaluate their method exclusively on vision and tabular datasets and do not cover text datasets. Suriyakumar et al. (2021) restrict their evaluation to tabular clinical data (MIMIC), and Tran et al. (2025) (“FairDP”) examine only vision and tabular datasets without including text. Across the field, it is standard for DP studies to concentrate on a subset of modalities rather than covering vision, text, tabular, and clinical domains simultaneously.
>
> On the choice of the smooth function (tanh):
>
> We selected the tanh function because it satisfies the key requirement for differential privacy that f(x)≤x  for x>0, ensuring that gradients are only reduced and never amplified. Alternative nonlinear functions such as sigmoid or softsign do not consistently meet this condition. Specifically, the standard sigmoid function does not start from zero, making it less suitable for controlling sensitivity. It can also produce outputs larger than the input magnitude near the origin, violating the suppression constraint, and it is known to saturate too early, which makes optimization more difficult (Glorot and Bengio, 2010). Although a shifted version (e.g., 2σ(x)−1) centers the function around zero, its curvature still causes it to exceed the identity line for small inputs and it saturates too early, which can make optimization less stable (Glorot and Bengio, 2010).
> In contrast, both tanh and softsign delay saturation relative to sigmoid (Elsayed et al., 2019). Among these, tanh remains the closest to the identity line for small values of x while still smoothly saturating toward 1, providing a stable transition that mimics hard clipping while preserving useful gradient information. This balance between gradient preservation and stability makes tanh the most appropriate choice for fairness-aware private optimization.
>
> [1] Xavier Glorot and Yoshua Bengio. Understanding the difficulty of training deep feedforward neural networks. In Proceedings of the thirteenth international conference on artificial intelligence and statistics, pages 249–256. JMLR Workshop and Conference Proceedings,2010.
>
> [2] Nelly Elsayed, Anthony Maida, and Magdy Bayoumi. Effects of different activation functions for unsupervised convolutional lstm spatiotemporal learning. Advances in Science, Technology and Engineering Systems Journal, 4(2):260–269, 2019.

---

### Official Review · Reviewer_3XfE · 2025-11-01

**Soundness:** 3
**Presentation:** 3
**Contribution:** 3
**Rating:** 4
**Confidence:** 4

**Summary:**

The paper provides a novel approach to building fair and private model training through adaptive clipping  using tanh transformation that preserves magnitudes of gradients. The paper addresses a clear problem of privacy-fairness tradeoff with a unique approach.

**Strengths:**

- Mechanism design is clean and nicely presented, can be integrated well into the existing libraries and pipelines
- tanh idea is also quite strong, from the literature I know DPSGD always focused on balancing noise multiplier with clipping bound without focusing on how the clipping is performed.
- Empirically there is some evidence of improved fairness with same guarantees

**Weaknesses:**

Overall, while the paper looks great it lacks experimental evidence of the usefulness of the method, here are a couple of questions:
- I believe the paper should include at least a basic example of applying proposed method to vision problems, even MNIST or CIFAR is enough. It is stated as out-of-scope in limitations, but it will still be very useful to have.
- Additionally unbalancing these vision datasets and demonstrating how far the method can apply.
- I would like to see how different epsilon changes the effect of the method.
- I am also curious about selection of the noise multiplier (noise/clipping threshold) and how this selection affects the method.
- Can we have different fairness metrics besides difference in loss? maybe accuracy on test sets?
- Comparison with related work. The paper only compares with DPSGD and adaptive clipping (Andrew et al) but even adaptive clipping has different settings. It will be helpful to also compare with Bu et al 2024 (automatic clipping).

**Questions:**

Addressing weakness above will significantly help the paper.

---

> ### Author Response · Authors · 2025-11-21
>
> On using a vision dataset (e.g., MNIST or CIFAR):
> We appreciate the reviewer’s suggestion. Our method operates at the optimizer and clipping level, which makes it inherently modality-agnostic. It can be applied to vision architectures without requiring any changes to the method itself. In this work, we focused on text-based clinical notes, tabular health data, and tabular demographic data. Our experimental design reflects common practice in the differential privacy literature, where evaluations typically center on one modality or a small number of related modalities rather than attempting to span all data types within a single paper. For example, Esipova et al. (2022) evaluate their method exclusively on vision and tabular datasets and do not cover text datasets. Suriyakumar et al. (2021) restrict their evaluation to tabular clinical data (MIMIC), and Tran et al. (2025) (“FairDP”) examine only vision and tabular datasets without including text. Across the field, it is standard for DP studies to concentrate on a subset of modalities rather than covering vision, text, tabular, and clinical domains simultaneously.
>
>
>
> On varying ε and noise multiplier:
> We thank the reviewer for raising this important point. In the revised manuscript, we have added clarification on how the noise multiplier is determined and how it interacts with the clipping threshold. Specifically, we now state:
> “As shown in prior work, smaller ϵ values tend to improve fairness (Tran et al., 2025) but reduce overall utility, whereas larger ϵ values provide stronger utility at the expense of privacy. The noise multiplier is computed jointly from ϵ, the sampling rate, and the clipping threshold using the RDP accountant, and its scale increases proportionally with the clipping norm. Following the literature, we adopt ϵ = 8 as a practical midpoint that offers a reasonable trade-off between privacy, fairness, and utility, which is why we use this value in all of our experiments.”
> This text has been added to the Experimental Setup section (in the paragraph describing training with DP-Adam and Opacus).
>
> On fairness metrics beyond loss difference:
> We appreciate the reviewer’s suggestion. In this work, we focused primarily on loss-based fairness metrics, as our goal was to analyze how gradient clipping affects the training dynamics and loss disparity across demographic subgroups. While metrics such as accuracy or F1-score are important for evaluating model performance, the loss metric provides a more direct view of the optimization process and fairness behaviour during training.
>
> On Comparisons With Additional Clipping Strategies like Bu et al. (2024):
> Thank you for the helpful suggestion. We did not include Bu et al. (2024) in our experiments because their method adopts a conceptually different approach, which is automatic clipping based on gradient length, whereas our work focuses on smooth adaptive transformations of gradient magnitudes. Unlike SoftAdaClip, which preserves gradient structure and applies a smooth transformation to suppress large gradients, Bu et al. (2024) aim to equalize per-sample gradient norms by automatically selecting a global clipping threshold. Because their method forces gradients toward a common length while ours is designed to preserve relative gradient magnitudes for fairness analysis, the two operate in fundamentally different regimes. Therefore, comparing them directly would conflate differing objectives and mechanisms. To ensure a fair comparison, we focused our baselines to methods that are most closely aligned with ours (DP-SGD and adaptive clipping by Andrew et al.).

---

### Meta-Review · Area_Chair_CQ3N · 2025-12-16

**Summary:**

This paper introduces a modification to the algorithm proposed by  Andrew et al. (2021). Building on the observation of Esipova et al. (2022) (that more aggressive clipping can benefit minority groups) the authors replace the (adaptive) hard‑clipping step, which uses a min operator, with a smooth clipping function based on the hyperbolic tangent (tanh). Empirical results demonstrate that this modification yields a better trade‑off among accuracy, fairness, and privacy.

The observation and motivation are sound and interesting. However, the paper is limited to an experimental study supported by a certain intuitive insight.  It has no theoretical analysis that justifies why fairness can be improved. Moreover, the empirical study lacks experimental evidence and some comparisons with alternative approaches.

**Reviewer Concerns:**

Most of the reviewers have pointed the necessity to make substantial changes to the experimental part of the paper, motivated by the fact that the main conclusions about fairness are empirical. A real effort has been made. However, some recommendations have been only partially taken into account (application to vision, fairness metrics, comparison to other methods).

**Reviewer Scores:**

Zq5M  might have increased his score to 6 because an effort has been made on some experiments. (Update: he revised his review but did not change his score.)

---

### Decision · Program_Chairs · 2026-01-26

Reject